# DNA requirement in FANCD2 deubiquitination by USP1-UAF1-RAD51AP1 in the Fanconi anemia DNA damage response

Fengshan Liang[1,2], Adam S. Miller[1], Simonne Longerich[1], Caroline Tang[2,3], David Maranon[4], Elizabeth A. Williamson[5], Robert Hromas[5], Claudia Wiese[4], Gary M. Kupfer[2,3] & Patrick Sung[1,6]

Fanconi anemia (FA) is a multigenic disease of bone marrow failure and cancer susceptibility stemming from a failure to remove DNA crosslinks and other chromosomal lesions. Within the FA DNA damage response pathway, DNA-dependent monoubiquitinaton of FANCD2 licenses downstream events, while timely FANCD2 deubiquitination serves to extinguish the response. Here, we show with reconstituted biochemical systems, which we developed, that efficient FANCD2 deubiquitination by the USP1-UAF1 complex is dependent on DNA and DNA binding by UAF1. Surprisingly, we find that the DNA binding activity of the UAF1-associated protein RAD51AP1 can substitute for that of UAF1 in FANCD2 deubiquitination in our biochemical system. We also reveal the importance of DNA binding by UAF1 and RAD51AP1 in FANCD2 deubiquitination in the cellular setting. Our results provide insights into a key step in the FA pathway and help define the multifaceted role of the USP1-UAF1-RAD51AP1 complex in DNA damage tolerance and genome repair.

---

[1] Department of Molecular Biophysics and Biochemistry, Yale University School of Medicine, New Haven, CT, USA. [2] Department of Pediatrics, Section of Hematology-Oncology, Yale University School of Medicine, New Haven, CT, USA. [3] Department of Pathology, Yale University School of Medicine, New Haven, CT, USA. [4] Department of Environmental and Radiological Health Sciences, Colorado State University, Fort Collins, CO, USA. [5] Department of Medicine, University of Texas Health Science Center at San Antonio, San Antonio, TX, USA. [6] Department of Biochemistry and Structural Biology, University of Texas Health Science Center at San Antonio, San Antonio, TX, USA. Correspondence and requests for materials should be addressed to G.M.K. (email: gary.kupfer@yale.edu) or to P.S. (email: sungp@uthscsa.edu)

Cells from patients with Fanconi anemia (FA) exhibit hypersensitivity to mitomycin C (MMC), reactive aldehydes, and other agents that cause interstrand DNA crosslinks (ICLs) or interfere with DNA replication[1,2]. Twenty two FA complementation groups, FA-A to FA-W, and their associated genes have been identified thus far[3,4]. In the execution of the FA DNA damage response, the E2 ubiquitin conjugating enzyme FANCT (UBE2T) works in conjunction with the E3 ubiquitin ligase FANCL[5,6] to monoubiquitinate FANCD2 while the FANCI-FANCD2 (ID2) heterodimer is bound to DNA[7–10]. The E3 ligase activity of FANCL is upregulated upon its association with FANCB and FAAP100 to form the BL100 complex[11–13].

The mono-ubiquitinated ID2 complex appears to license downstream events of DNA damage repair or tolerance. Specifically, during ICL repair, mono-ubiquitinated ID2 is thought to recruit homologous recombination factors, such as the tumor suppressor BRCA2 and the recombinase enzyme RAD51, to eliminate the DNA double-strand break (DSB) arising from the nucleolytic processing of the DNA crosslink[14–16]. Likewise, mono-ubiquitinated ID2 co-ordinates the replicative bypass of unhooked DNA crosslinks by the translesion DNA synthesis polymerase ζ[17–19].

Upon DNA lesion removal or bypass, timely deubiquitination of FANCD2 becomes critical for extinguishing the DNA damage response and for regenerating the ID2 complex[20–22]. This critical step is mediated by the deubiquitinating enzyme (DUB) USP1 in complex with UAF1, which serves as the ID2 targeting subunit of the DUB complex. USP1-UAF1 also acts on the mono-ubiquitinated form of the DNA polymerase processivity factor PCNA in translesion DNA synthesis (TLS)[23]. Notably, UAF1 partners with two other DUBs, USP12 and USP46, in the removal of ubiquitin from other cellular targets[24–26]. Interestingly, UAF1 binds DNA and participates in DSB repair by homologous recombination[21,27,28]. Owing to the broader involvement of UAF1 in DNA damage repair and other biological processes, its loss engenders a more severe phenotype than USP1 ablation, as evidenced by early embryonic lethality in mice and a higher degree of cellular sensitivity to DNA damaging agents[20,21,28,29].

Both the unmodified and mono-ubiquitinated forms of ID2 bind DNA avidly, and there is good evidence that modified ID2 remains associated with DNA lesions until damage repair or translesion DNA synthesis has occurred[10]. Importantly, through in vitro reconstitution with highly purified proteins, it has been demonstrated that FANCD2 monoubiquitination occurs efficiently only when ID2 is bound to DNA[9,30]. However, it was suggested recently that DNA shields ubiquitinated ID2 from the DUB activity of USP1-UAF1, such that the removal of DNA would lead to a marked enhancement of the efficiency of ID2 deubiquitination[11].

We wish to define mechanistically how DNA may affect the efficiency of FANCD2 deubiquitination by USP1-UAF1, and have developed a reconstituted system of highly purified human proteins for this purpose. Surprisingly, and in direct contradiction with the recently published report[11], we find that the deubiquitination of FANCD2 by USP1-UAF1 requires DNA being present. Consistent with this finding, we provide evidence that the DNA binding activity of UAF1 is indispensable for FANCD2 deubiquitination. Interestingly, the FANCD2 DUB activity of USP1-UAF1 that harbors a DNA binding defective UAF1 mutant is fully restored upon the incorporation of wild type but not a DNA binding defective form of RAD51AP1. In congruence with the above observations, we show that simultaneously inactivating the DNA binding attribute of UAF1 and RAD51AP1 engenders a deficiency in FANCD2 deubiquitination in cells. These results help establish that monoubiquitination and deubiquitination of FANCD2 occur sequentially on DNA, implicate RAD51AP1 in the FA pathway of DNA damage response, and identify the functions that the DNA binding activity of UAF1 and RAD51AP1 fulfill in the FA pathway of DNA damage response. Moreover, our results suggest the intriguing possibility that RAD51AP1-UAF1 may similarly act in partnership with other DUBs to deubiquitinate substrates on DNA.

## Results

**Factors for FANCD2 ubiquitination and deubiquitination.** Wild type and mutant forms of RAD51AP1, the ID2 complex, and the USP1-UAF1 complex were expressed and purified as described[9,27,31,32] (Supplementary Fig. 1a). We used the multibac described by van Twest et al.[11] to express the BL100 complex in insect cells and devised a method for its purification (Supplementary Fig. 1a). We have also purified BL100 from insect cells co-infected with three separate recombinant baculoviruses that we constructed, but found no difference in specific activity as compared to BL100 purified from the multibac system.

**Requirement for DNA in FANCD2 deubiquitination.** Given that ID2 mono-ubiquitination requires DNA[9,30] and that modified ID2 remains associated with chromatin in cells[10], we hypothesized that ID2 deubiquitination may also be dependent on DNA. This premise was tested in a reconstituted system that we have developed, as below.

To generate mono-ubiquitinated FANCD2, we incubated the ID2 complex with ATP, DNA, UBE1, UBE2T/FANCT, BL100 complex, and HA-ubiquitin. Under our reaction conditions, ~60% of FANCD2 became monoubiquitinated within 30 min of incubation (Fig. 1a, b, lane 2). To test the DUB activity of USP1-UAF1, we first depleted the ATP content (to prevent further FANCD2 ubiquitination) of the above reaction by apyrase treatment. Efficient deubiquitination of FANCD2 occurred upon the addition of USP1-UAF1 (Fig. 1a, b). To ask whether DNA is needed for deubiquitination, we treated the reaction mixture with benzonase to digest the DNA before adding USP1-UAF1 (Fig. 1b, lanes 5, 7, 9, and 11; Supplementary Fig. 1b, lanes 4–7). Importantly, DNA removal reproducibly led to a marked decrease in FANCD2 deubiquitination (≥95% inhibition). The same degree of inhibition of FANCD2 deubiquitination was seen when we used DNase I to remove DNA (Fig. 1c, lane 7). We also employed Ub-VS as activity probe[33] to establish that neither benzonase nor DNase I affects the intrinsic DUB attribute of USP1-UAF1 (Supplementary Fig. 1c). Taken together, the results above helped establish that FANCD2 deubiquitination requires DNA being present. Consistent with this deduction, we present evidence below to implicate the DNA binding activity of UAF1 in FANCD2 deubiquitination.

**Construction of UAF1 DNA binding mutants.** To determine whether the DNA binding activity of UAF1[27] is needed for FANCD2 deubiquitination, we strived to isolate mutants that are deficient in this UAF1 attribute. As reported before and recapitulated in Fig. 2, we found by DNA electrophoretic mobility shift that UAF1[436X] (which harbors residues 1–436 and all the WD40 repeats of UAF1) binds DNA just as avidly as full-length UAF1, whereas UAF1[SLD] (which contains residues 390–677 and the tandem SLD repeats) is devoid of such activity (Fig. 2a, b, Supplementary Fig. 2a and Supplementary Table 3).

We note that UAF1 resembles DDB2, which functions in nucleotide excision repair, in possessing DNA binding activity within its WD40 repeats[34]. As revealed by X-ray crystallography of the *Danio rerio* ortholog, DDB2 employs a number of solvent-exposed basic and other residues to engage its DNA ligand[34]. To

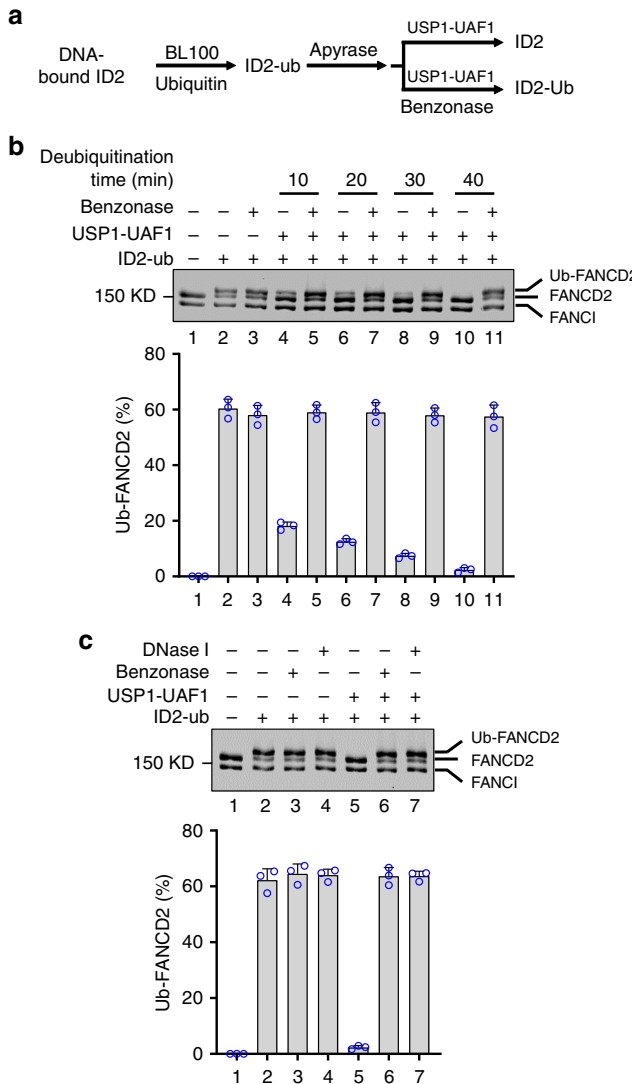

**Fig. 1** Requirement for DNA in FANCD2 deubiquitination. **a** Schematic of ID2 ubiquitination and deubiquitination in our reconstituted system. **b** ID2 was first incubated with BL100 to ubiquitinate FANCD2, followed by treatment with USP1-UAF1 without or with benzonase. The error bars represent the mean+S.D. of data from three independent experiments. **c** Either benzonase or DNase I treatment abolished FANCD2 deubiquitination by USP1-UAF1. The error bars represent the mean+S.D. of data from three independent experiments

help guide our effort to isolate UAF1 DNA binding mutants, we used the molecular visualization programs CCP4 (http://www.ccp4.ac.uk)[35] to superimpose the crystal structure of the WD40 repeats of DDB2 (chain D) with UAF1[36]. Based on further analysis with the ConSeq neural-network algorithm (http://consurf.tau.ac.il/)[37], eleven amino acid residues within the WD40 repeats of UAF1 were selected as mutagenesis targets. Two mutants were constructed, with one harboring the change of three of the selected residues to alanine (the UAF1[3A] mutant) and the other with all eleven residues having been changed to alanine (the UAF1[11A] mutant; see Experimental Procedures, Supplementary Tables 1, 3 and Supplementary Fig. 2a, b for details).

Both the UAF1[3A] and UAF1[11A] mutants could be expressed and purified to a high degree following procedures developed for the wild-type counterpart (Supplementary Fig. 2a). Importantly, we found by testing three different substrates (ssDNA, dsDNA,

and D-loop) that the UAF1[3A] and UAF1[11A] mutants are deficient in DNA binding (Fig. 2c, d and Supplementary Table 3). Importantly, we have verified that UAF1[3A] and UAF1[11A] retain the ability to interact with USP1, RAD51AP1, and FANCI (Supplementary Fig. 2c–e) and, with the Ub-VS DUB assay, to stimulate the DUB activity of USP1 (Supplementary Fig. 2f).

**Requirement for UAF1 DNA binding in FANCD2 deubiquitination.** To ask if UAF1 DNA-binding is relevant for ID2 deubiquitination, we examined the mutant USP1-UAF1[3A] and USP1-UAF1[11A] complexes (Supplementary Fig. 1a) in FANCD2 deubiquitination. Importantly, the results revealed that neither mutant complex is able to catalyze FANCD2 deubiquitination (Fig. 3a, lanes 5 and 7) even though, as documented above and in Supplementary Fig. 2c–f, both UAF1 mutants retain all other known biochemical attributes, namely, interactions with RAD51AP1, USP1, and FANCI, and also the ability to enhance the DUB activity of USP1. We conclude that the DNA binding activity of UAF1 is indispensable for FANCD2 deubiquitination in our reconstituted biochemical system.

**Role of RAD51AP1 DNA binding in FANCD2 deubiquitination.** Since RAD51AP1, which stably associates with UAF1[27], also possesses DNA binding activity[32], it was of considerable interest to test whether it would restore to USP1-UAF1[3A] and USP1-UAF1[11A] mutant complexes the ability to deubiquitinate FANCD2 on DNA. Importantly, the addition of RAD51AP1 to reactions containing either USP1-UAF1[3A] or USP1-UAF1[11A] led to robust FANCD2 deubiquitination (Fig. 3b, lanes 7 and 9; Supplementary Fig. 3a). As expected, RAD51AP1 and USP1 together failed to deubiquitinate FANCD2 when UAF1 was absent (Supplementary Fig. 3b, c).

We had previously isolated a RAD51AP1 variant, referred to as RAD51AP1[DM], that lacks DNA binding activity (ref. [31]; Supplementary Table 3 and Supplementary Fig. 3d). Notably, even though RAD51AP1[DM] could associate with UAF1 and the UAF1[3A] and UAF1[11A] mutants just as avidly as its wild type counterpart (Supplementary Fig. 3e and ref. [27].), it failed to restore FANCD2 DUB activity to USP1-UAF1[3A] (Fig. 3b, lanes 7 and 11). Taken together, the results above indicate that the DNA binding activity of RAD51AP1 can substitute for that of UAF1 in FANCD2 deubiquitination.

**Cellular role of UAF1 and RAD51AP1 DNA binding.** To interrogate the biological relevance of the UAF1 DNA binding activity, we generated a HeLa cell line constitutively expressing shRNA targeting the 3′UTR of UAF1 and transfected the UAF1-depleted cells with cDNA encoding either the wild type or the 11A or 3A mutant form of UAF1 (Fig. 4a). As expected, UAF1 depletion led to hypersensitivity of cells to MMC and olaparib, the latter being an inhibitor of polyADP-ribose polymerase 1 (PARP1) (Fig. 4b). Complementation with UAF1 rendered cells resistant to these agents, but neither the UAF1[3A] nor the UAF1[11A] mutant could fully restore resistance (Fig. 4b), even though these mutants were expressed to the same level as the wild-type protein (Fig. 4a). Thus, the DNA binding activity of UAF1 is needed for chromosome damage repair. Interestingly, whereas, as expected[24,27,28], UAF1 deficient cells showed an elevated level of monoubiquitinated FANCD2, but cells expressing either the UAF1[3A] or UAF1[11A] mutant harbored mostly deubiquitinated FANCD2 (Fig. 4a), indicating that the UAF1 mutations have little or no impact on the FANCD2-specific DUB activity of the USP1-UAF1 complex in cells.

Our biochemical reconstitution studies (Fig. 3b and Supplementary Fig. 3a) revealed that, via its DNA binding activity,

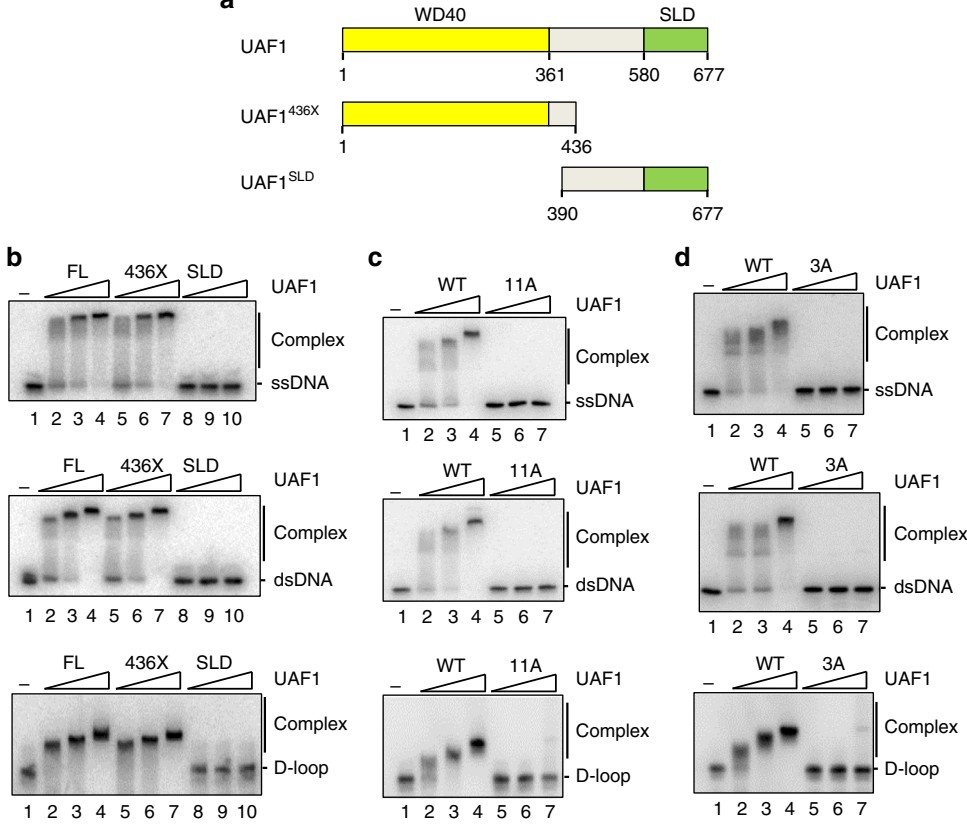

**Fig. 2** Biochemical analysis of UAF1 DNA binding mutants. **a** UAF1 fragments tested for DNA binding. Numbers indicate the positions of amino acid residues. **b** UAF1 (FL), UAF1[436X] and UAF1[SLD] (100, 200 and 400 nM) were tested for the binding of ssDNA, dsDNA, or D-loop (10 nM). **c** UAF1 (WT) and UAF1[11A] (11A) (200, 400, and 800 nM) were tested for the binding of ssDNA, dsDNA, or D-loop as in **b**. **d** UAF1 (WT) and UAF1[3A] (3 A) were tested for the binding of ssDNA, dsDNA or D-loop as in **c**

RAD51AP1 targets the DNA binding defective USP1-UAF1[3A] or USP1-UAF1[11A] complex to deubiquitinate FANCD2 on DNA. Given this, we wished to ask whether simultaneous loss of DNA binding in UAF1 and RAD51AP1 would have a negative impact on FANCD2 deubiquitination in cells. As shown in Fig. 4c and Supplementary Fig. 4a, HeLa cells that are deleted for *RAD51AP1* are proficient in FANCD2 deubiquitination even though, as expected[32], they exhibit hypersensitivity to MMC (Supplementary Fig. 4b). In these RAD51AP1 knockout cells, siRNA-mediated knockdown of UAF1 also led to elevated levels of ubiquitinated FANCD2 and introduction of UAF1 restored FANCD2 deubiquitination (Fig. 4c, Lanes 3 and 4). Importantly, RAD51AP1 null cells expressing the UAF1[3A] or UAF1[11A] mutant harbored an abnormally high level of ubiquitinated FANCD2 and also showed increased MMC sensitivity, with the UAF1[11A] mutant exhibiting the more severe phenotype in these regards (Fig. 4c, lanes 7 and 10; Fig. 4d). While the introduction of wild-type RAD51AP1 into these cells restored the ability to deubiquitinate FANCD2, expression of the RAD51AP1[DM] mutant did not (Fig. 4c, lanes 8, 9, 11, and 12). Thus, in congruence with our biochemical results, the genetic analysis provides evidence for a redundant role of the DNA binding activities of UAF1 and RAD51AP1 in the mediation of FANCD2 deubiquitination.

## Discussion
UAF1 has been regarded as the substrate recognition subunit of the USP1-UAF1 complex[38,39]. Importantly, we have documented a function of the UAF1 in conjunction with RAD51AP1 in DNA damage repair as well[27]. In this study, we have employed reconstitution biochemistry and cell-based tests to define the importance of the DNA binding attribute of UAF1 and RAD51AP1 in FANCD2 deubiquitination (Fig. 4e). Several findings have emerged, namely: (1) occurrence of USP1-UAF1 mediated-deubiquitination of FANCD2 on DNA; (2) identification of UAF1 amino acid residues important for DNA binding activity; (3) demonstration that UAF1 DNA binding as being required for FANCD2 deubiquitination within the context of the USP1-UAF1 complex; (4) documentation that DNA binding by RAD51AP1 can substitute for that of UAF1 in FANCD2 deubiquitination within the context of the trimeric USP1-UAF1-RAD51AP1 complex; and (5) revelation that UAF1 and RAD51AP1 provide redundant functions for FANCD2 deubiquitination in cells, thus uncovering a role of RAD51AP1 in the FA DNA damage response in addition to its well-documented DNA repair function.

We note that a main conclusion of our study concerning the indispensable role of DNA in FANCD2 deubiquitination contradicts what was reached by van Twest et al.[11], who suggested that FANCD2 deubiquitination occurs much more efficiently when mono-ubiquitinated ID2 has dissociated from DNA than when it is DNA-bound. In that study[11], the human FA core complex was used to ubiquitinate the FancI-FancD2 complex from *Xenopus laveis*, and the modified *Xenopus* ID2 complex was then tested with human USP1-UAF1. Given that human FANCI and FANCD2 share relatively low sequence identity with their *Xenopus* counterparts (~65% for FANCI and ~55% for FANCD2), it seems plausible that the requirement of DNA for efficient FANCD2 deubiquitination by human USP1-UAF1 becomes relaxed with the *Xenopus* protein. Alternatively, although less likely in our view, FancD2 deubiquitination in

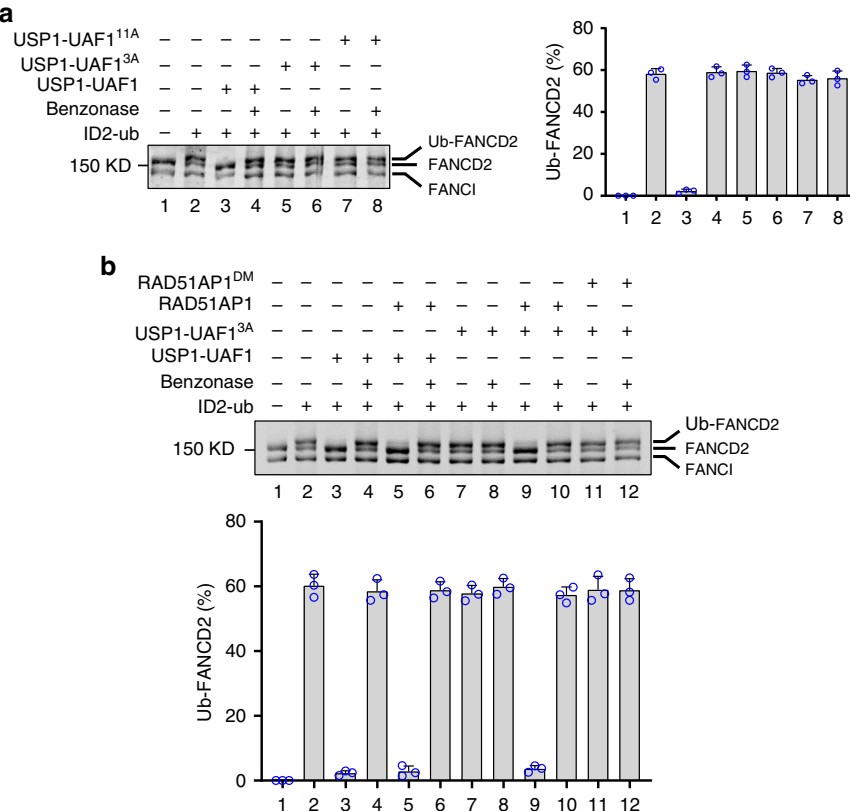

**Fig. 3** Role of UAF1 and RAD51AP1 DNA binding in FANCD2 deubiquitination. **a** Deubiquitination of FANCD2 by USP1-UAF1 that harbored wild type or mutant UAF1. The error bars represent the mean+S.D. of data from three independent experiments. **b** Testing of RAD51AP1 and RAD51AP1$^{DM}$ in FANCD2 deubiquitination. The error bars represent the mean+S.D. of data from three independent experiments

*Xenopus laevis* may occur after monoubiquitinated ID2 has dissociated from DNA.

We have shown that cells deleted for RAD51AP1 are proficient in FANCD2 ubiquitination and deubiquitination but hypersensitive to MMC (Supplementary Fig. 4). The drug sensitivity seen in this study and elsewhere can be attributed to an impairment of DNA damage repair by homologous recombination[27,32,40], a distinct attribute of RAD51AP1 in addition to its involvement in FANCD2 deubiquitination (this study).

We reported previously that the DNA binding function of RAD51AP1 is indispensable for homology-directed DNA damage repair (HDR)[31]. Importantly, evidence has been presented herein that the UAF1 DNA binding activity is also needed for cellular resistance to MMC and olaparib while being dispensable for FANCD2 deubiquitination in cells. Thus, even though DNA binding by either UAF1 or RAD51AP1 is sufficient for targeting the USP1-UAF1-RAD51AP1 DUB complex to DNA for FANCD2 deubiquitination, the DNA binding attribute of both UAF1 and RAD51AP1 is required for optimal DNA damage repair.

Recently, Lim et al.[41], reported that USP1 also possesses a DNA binding activity with specificity for forked DNA (i.e. Y-shaped DNA with a duplex arm) and provided evidence that DNA binding exerts a modest stimulation of the DUB activity of USP1-UAF1. Furthermore, BRCA1-deficient cells expressing a truncated variant of USP1 deficient in DNA binding fail to deliver USP1-UAF1 to replication forks to act on the ubiquitinated form of the proliferating cell nuclear antigen (PCNA) and are, consequently, impaired for the ability to stabilize stressed DNA replication forks[41]. It will be of considerable interest to determine whether the DNA binding activity of UAF1 and RAD51AP1 is also needed for the preservation of stressed replication forks and if DNA binding by USP1 is germane for FANCD2 deubiquitination and HDR proficiency.

The experimental systems that we and others[11,41] have developed will be useful for answering mechanistic questions regarding the intracellular functions of the USP1-UAF1 and for establishing the biological impact of cancer-associated mutations in the USP1-UAF1-RAD51AP1 complex. Components of the FA DNA damage response pathway represent promising therapeutic targets in HDR-deficient tumors, such as those caused by BRCA1 inactivation[41,42]. In this regard, chemical compounds that target protein–protein and DNA binding interfaces within the USP1-UAF1-RAD51AP1 complex would be useful not only as a chemical biology tool, but could also be further developed into cancer therapeutics.

## Methods

**Proteins and protein complexes**. A procedure entailing affinity, ion exchange, and gel filtration steps was used to purify human UAF1$^{436X}$, UAF1$^{SLD}$, UBE2T, the USP1-UAF1 complex, and the ID2 complex[9,27]. RAD51AP1 and the RAD51AP1$^{DM}$ mutant were expressed in *E. coli* and purified by GST or MBP tag affinity chromatography and Source S (GE HealthCare) ion exchange chromatography[31,32]. FANCB-FANCL-FAAP100 (BL100) complex with N-terminally Flag-tagged FANCB was expressed in insect cells using the multi-bacmid provided by Andrew Deans (University of Melbourne, Australia) and purified by Flag tag affinity chromatography and MiniQ (GE Healthcare) ion exchange chromatography[11], with an added terminal step of size exclusion in Superdex 200 using buffer A (20 mM Tris-HCl, pH 7.5, 10% glycerol, 0.5 mM EDTA, 0.01% Igepal, 1 mM DTT, and 150 mM KCl). HA-Ubiquitin and UBE1 were purchased from Boston Biochem.

**Construction of UAF1 mutants**. The cDNAs for human UAF1 (isoform 1) and mutant derivatives were introduced into the pFastBac-1 (Thermo Fisher) vector containing a C-terminal Strep-II tag. The UAF1$^{3A}$ mutant (see Supplemental Table 3) was generated using QuikChange site-directed mutagenesis (Agilent) with oligonucleotides listed in Supplemental Table 1. The UAF1$^{11A}$ mutant gene (see

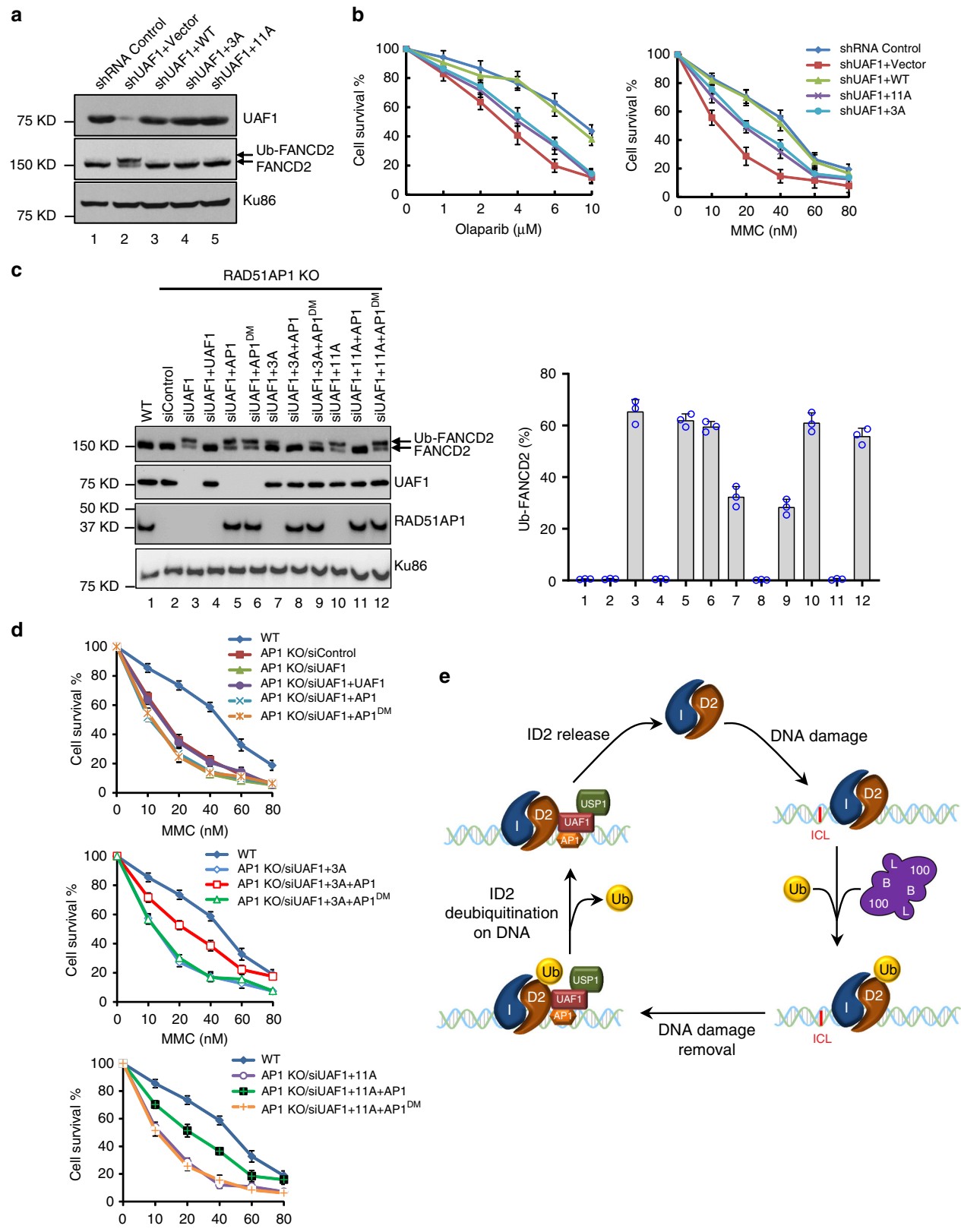

Supplemental Table 3) was synthesized by Genewiz. Bacmids were generated in *E. coli* strain DH10Bac.

**Purification of UAF1 and mutants**. The Bac-to-Bac TOPO expression system (Thermo Fisher) was used to generate recombinant baculoviruses in *Spodoptera frugiperda* (Sf9) cells (Invitrogen). Viruses were used to infect HighFive insect cells

(Invitrogen) at a density of $1 \times 10^6$ cells/ml. After a 48-hour incubation at 27 °C, cells were collected by centrifugation and stored at −80 °C. All purification steps were performed at 0–4 °C. A 6-g cell pellet (from 400 ml of insect cell culture) was thawed in 50 ml cell breaking buffer (20 mM Tris-HCl, pH 7.5, 10% glycerol, 0.5 mM EDTA, 300 mM KCl, 0.1% Igepal CA-630, 1 mM DTT, 1 mM PSMF, 3 mM β-mercaptoethanol (BME), and a cocktail of protease inhibitors comprising aprotinin, chymostatin, leupeptin, and pepstatin A at 5 μg/ml each), and cells were

**Fig. 4** Role of UAF1 and RAD51AP1 in FANCD2 deubiquitination in cells. **a** HeLa cells depleted for endogenous UAF1 and stably expressing UAF1 (WT), UAF1[3A] (3A), or UAF1[11A] (11A) were examined for their level of ubiquitinated FANCD2 by western blotting. Ku86 was included as the loading control. **b** Survival of cells (from **a**) upon treatment with MMC or olaparib. The error bars represent mean values±S.D. of data from three independent experiments. **c** RAD51AP1 knockout HeLa cells depleted for endogenous UAF1 and stably expressing WT or mutant UAF1 with or without the co-expression of either RAD51AP1 (AP1) or the RAD51AP1[DM] (DM) mutant were examined for their level of ubiquitinated FANCD2. Ku86 was included as the loading control. The error bars represent the mean+S.D. of data from three independent experiments. **d** Survival of cells (from **c**) upon treatment with MMC. The error bars represent mean values±S.D. of data from three independent experiments. **e** Model for the role of the USP1-UAF1-RAD51AP1 complex in the FA pathway. Mono-ubiquitination and deubiquitination of FANCD2 both occur on DNA. DNA binding by either UAF1 or RAD51AP1 is sufficient for targeting the trimeric DUB complex to DNA bound, mono-ubiquitinated FANCD2

ruptured by sonication with a Bransen Sonifier on ice (three 30 s pulses at 50% output, setting 5). The crude lysate was clarified by ultracentrifugation (100,000×g, 75 min) and then diluted with 3 volumes of T buffer (20 mM Tris-HCl, pH 7.5, 10% glycerol, 0.5 mM EDTA, 0.01% Igepal CA-630, 3 mM BME, and 1 mM DTT). The diluted lysate was applied onto a 25-ml column of Q Sepharose fast flow (Amersham Biosciences), which was washed with 150 ml of T buffer with 50 mM KCl and then developed using a 200-ml gradient of 50–500 mM KCl in T buffer, collecting 5 ml fractions. Fractions containing the peak of UAF1 (220–300 mM KCl) were combined and passed through a 1-ml column of streptactin affinity matrix (IBA), and, after washing the column with 25 ml each of T buffer containing 1000 mM KCl and T-buffer containing 300 mM KCl, bound proteins were eluted with 10 ml of T-buffer containing 300 mM KCl and 2.5 mM desthiobiotin in 1 ml fractions. Fractions 2–6 were pooled and concentrated to 500 µl (Ultracel-30K concentrator from Amicon) and then fractionated in a Superdex 200 gel filtration column (24 ml bed volume) in T buffer containing 300 mM KCl and 1 mM DTT, collecting 0.5 ml fractions. Highly purified UAF1 and mutants were concentrated to ~1.5 mg/ml, frozen in liquid nitrogen in 2 µl aliquots, and stored at −80 °C.

**Affinity pull-down and DNA mobility shift assay**. To test interaction of UAF1 with RAD51AP1, GST or MBP-tagged RAD51AP1 (2 µg) and Strep II-tagged UAF1 (2 µg) were incubated in 30 µl reaction buffer (25 mM Tris-HCl, pH 7.5, 10% glycerol, 0.5 mM EDTA, 0.01% Igepal, 1 mM DTT, and 150 mM KCl) for 30 min. Then, the reaction was mixed gently with 10 µl glutathione resin (GE Healthcare) or amylose resin (New England Biolabs) at 4 °C for 1 h, which was collected by centrifugation, washed with 50 µl buffer, and treated with 30 µl of 2% SDS to elute bound proteins. The supernatant, wash, and SDS eluate (10 µl each) were analyzed by 8% SDS-PAGE and Coomassie Blue staining. The same procedure was followed in testing UAF1-USP1 and UAF1-FANCI interactions, except that Strep-Tactin resin was used to capture protein complexes via the Strep tag on UAF1[27]. For DNA binding, the single-stranded DNA substrate was [32]P-labeled oligonucleotide P1 (Supplementary Table 2). The dsDNA and D-loop substrates were generated using oligonucleotides P1/P2 (P1 was radiolabeled) and oligonucleotides D1/D2/D3 (D3 was radiolabeled), respectively (Supplementary Table 2).

**Cell culture and transfection**. HeLa cells (ATCC) were cultured in DMEM medium (Gibco) with 10% FBS (ATLANTA Biologicals) and 1% Pen-Strep (Gibco). The shRNA in pSUPER.retro.puro vector and siRNA targeting the 3′ UTR of UAF1 have been described[27]. FLAG-tagged cDNAs coding for the wild type and mutant forms of UAF1 or RAD51AP1 were introduced into the pCMV(delta4) vector[43]. Plasmids were transfected into cells using Lipofectamine 2000 (Invitrogen). RAD51AP1 knockout HeLa cells were generated by transfecting HeLa cells with a cocktail of three different CRISPR/Cas9 knockout plasmids (Santa Cruz Biotechnology; sc-408187) each encoding Cas9 nuclease and one of three different RAD51AP1-specific 20 nucleotide gRNAs (targeting exons 2, 3, or 5/6). The sense sequences for these gRNAs were as follows: TTTGACCACTCTGACAGTGA (exon 2), AACCTAACTTGAACAATCTC (exon 3), and AGTGTAGCCAGTGAT-TATTT (exon 5/6). Cells were trypsinized 48 h post transfection and seeded in 96-well plates at 1 cell/ml to obtain clonal isolates. Loss of RAD51AP1 expression was confirmed by both western blot analysis and immunocytohistochemistry.

**Western blotting**. Western blots were probed with anti-WDR48 (1:1000, PA5–24007, Thermo Fisher), anti-RAD51AP1 (1:500, ab88370, Abcam), anti-USP1 (1:500, ab108104, Abcam), anti-FANCD2 (1:1000, sc-20022, FI17, Santa Cruz Biotechnology), or anti-Ku86 (1:5000, sc-515736, B-4, Santa Cruz Biotechnology) antibody. After incubation with the secondary antibody (1:5000, NA934, NXA931, GE Healthcare), blots were developed using the SuperSignal ECL substrate (34580, Thermo Fisher).

**Cell survival assay**. Cell survival was tested by crystal violet assay. Briefly, cells were seeded into 6-well plates at $1 \times 10^4$ cells/well and incubated at 37 °C for 24 h. After the addition of MMC (Sigma) or olaparib (Selleck Chemicals), cells were incubated at 37 °C for 5 days. Surviving cells were fixed and stained with crystal violet dye, and absorbance was measured in a microplate reader (BioTek).

**Assay for DUB activity**. The DUB attribute of USP1-UAF1 activity was assessed using Ubiquitin Vinyl Sulfone (Ub-VS) as activity probe. Briefly, 5 µM Ub-VS (U-202, Boston Biochem) was incubated with 0.5 µM USP1, RAD51AP1, or USP1-UAF1 complex in 30 µl reaction buffer (50 mM Tris-HCl, pH 8.5, 50 mM NaCl, 1 mM DTT) at 30 °C for 2 h. USP1, UAF1, and RAD51AP1 were detected by western blot analysis. Benzonase or DNase I (0.2U each) were added to some reactions, as indicated.

**Ubiquitination and deubiquitination assays**. For FANCD2 ubiquitination, purified human ID2 complex (0.2 µM) was mixed with 64-mer ssDNA (0.2 µM, Supplementary Table 2) in the reaction buffer (50 mM Tris-HCl, pH 7.5, 25 mM KCl, 4 mM MgCl₂, 2 mM ATP, 0.01 mg/ml bovine serum albumin, 0.5 mM DTT) and incubated at 25 °C for 10 min, followed by the addition of 0.1 µM UBE1 (Boston Biochem), 0.3 µM UBE2T, 0.2 µM BL100, and 32 µM HA-ubiquitin (Boston Biochem) and a 30-min incubation at 25 °C in a final volume of 12.5 µl. In order to deplete the ATP content of the reaction mixtures, they were incubated with 0.5 U apyrase (New England Biolabs) for 10 min at 25 °C. To digest DNA, 0.2 U benzonase (MilliporeSigma) or DNase I (New England Biolabs) was added together with apyrase. For FANCD2 deubiquitination, ATP-free reaction mixtures containing ubiquitinated FANCD2 were incubated with 0.2 µM USP1-UAF1, 0.2 µM RAD51AP1, or the combination of the two protein species in a final volume of 17 µl at 25 °C for 40 min. After mixing with 6 µl of 8% SDS, 240 mM Tris-HCl, pH 6.8, 40% glycerol, 5% BME, and 0.04% bromophenol blue, reaction mixtures were analyzed by 6.5% SDS-PAGE. Gels were stained with Oriole Fluorescent Gel stain (BioRad) and analyzed in a gel documentation station (BioRad).

**Reporting summary**. Further information on research design is available in the Nature Research Reporting Summary linked to this article.

## Data availability

The data that support the findings of the study are available from the corresponding authors upon reasonable request.

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

## Acknowledgements

We thank Andrew Deans for providing the multi-bacmid for expressing the BL100 complex. This work was supported by CPRIT REI Award RR180029 and US National Institutes of Health grants R01 CA168635, U54 DK106857, R01 CA220123, R01 CA205224, R01 ES007061, R01 GM109645, P01 CA092584, and P30 CA054174. P.S. is the holder of the Robert A. Welch Distinguished Chair in Chemistry (AQ-0012).

## Author contributions

P.S., G.M.K., and F.L. conceived the study. F.L., A.S.M., C.W., G.M.K., and P.S. designed the experiments and analyzed the data. F.L., A.S.M., S.L., C.T., D.M., and E.A.W generated the materials and executed the experiments. F.L., A.S.M., C.W., R.H., G.M.K., and P.S. wrote the paper.

## Additional information

**Competing interests:** The authors declare no competing interests.

