## [Peer Review File · Nature Communications]

REVIEWERS' COMMENTS:

Reviewer #1 (Remarks to the Author):

The key licensing step in the Fanconi anemia pathway is ubiquitination of the of FANCD2. This modification is reversible. Therefore, understanding the mechanisms that affect the steady state levels (both ubiquitination and deubiquitinating) of ubiquitinated FANCD2 are important.

Here, Liang et al. use a powerful set of biochemical tools to reconstitute deubiquitination of FANCD2 by recombinant USP1-UAF1. This allowed to make several important conclusions, including:

- Deubiquitination occurs on DNA: while this is in disagreement with a published report, the authors conclusions are very convincing (see also below);
- DNA binding activity of UAF1 is important for deubiquitination (the authors were able to identify specific mutants that affect this capacity);
- RAD51AP1's DNA binding activity can compensate for defects in UAF1 DNA binding

The redundancy of UAF1's and RAD51AP's (and their physiological relevance) was demonstrated by cellular knockdown-rescue experiments.

This is a very well-done work, the experimental quality is very high, presentation is clear and data clearly support conclusions: there is very little else to do!

Only a few little things:

- page 3: correct "BL00"
- page 4: "both UAF1 mutants retain all other known biochemical activities", please specify in text
- Figure 4: the graphs are hard to follow, please make better color coding or split into more panels (specifically panel d)

Reviewer #2 (Remarks to the Author):

This is an interesting and well executed study that rewrites current views of Fanc D2-I deubiquitination by the UAF1-USP1 complex. Prior work from van Twest and Deans (Mol Cell 2017) showed that deubiquitination occurred more efficiently in the absence of DNA thus supporting a model whereby removal occurred prior to DUB activity. In contrast Liang and Sung show that DNA binding by UAF1 and RAD51AP1 are required for DUB removal of UB from D2. The data is derived from clean reconstitution systems, point mutations that disrupt binding, and congruent findings in cellular systems following knockdown and reconstitution. These findings change current paradigms for ubiquitination-deubiquitination cycles for D2-I and add a new understanding to the interaction of UAF1-USP1 with RAD51AP1. It is of broad interest to the DNA repair field and represents a significant contribution. There are a few minor issues that should be addressed prior to publication.

Specific points.

The authors note that RAD51AP1 loss allows D2-I deubiquitination despite rendering cells MMC hypersensitive. How does this fit with the model? It suggests that AP1 has additional roles, which should be more clearly discussed. How does AP1 affect responses to MMC other than D2-I-Ub?

Point-by-point response to the reviewer's critiques (NCOMMS-19-07572-T)

Reviewer #1:

1.page 3: correct "BL00"

Our response: We have corrected this to "BL100".

2. page 4: "both UAF1 mutants retain all other known biochemical activities", please specify in text

Our response: We have specified in the revised manuscript which of the known biochemical activities are retained in the two UAF1 mutants (page 4).

3. Figure 4: the graphs are hard to follow, please make better color coding or split into more panels (specifically panel d)

Our response: We have color-coded the survival curves in Figure 4b, and have also split Figure 4d into three color-coded panels. We believe these changes make it much easier to follow what information is being shown in the figures. We are grateful to the Reviewer for the suggestion.

Reviewer #2:

RAD51AP1 loss allows D2-I deubiquitination despite rendering cells MMC hypersensitive. How does this fit with the model? It suggests that AP1 has additional roles, which should be more clearly discussed. How does AP1 affect responses to MMC other than D2-I-Ub?

Our response: The reviewer highlighted a most interesting point regarding the multifaceted role that RAD51AP1 fulfills in the DNA damage response. In this regard, previous studies by us and others have provided evidence that RAD51AP1 plays an important role in DNA repair by homologous recombination (Modesti et al, Mol. Cell, 2007; Wiese et al, Mol. Cell, 2007; Liang, et al., Cell Rep. 2016). Importantly, by constructing and analyzing single and double mutants of RAD51AP1 and UAF1, we have unveiled a Fanconi anemia specific function of RAD51AP1 in the deubiquitination of FANCD2. We have added a paragraph in the Discussion (page 6) to highlight the dual role that RAD51AP1 fulfills in DNA break repair and the Fanconi anemia pathway of DNA damage response.